# Risk Coupling Evaluation of Social Stability of Major Engineering Based on N-K Model

Hongyan Yan [1], Zhouwei Zheng [1], Hanjie Huang [2], Xinyi Zhou [3], Yizhi Tang [1] and Ping Hu [1,4,*]

1   Department of Construction Management, Hunan University of Finance and Economics,
    Changsha 410205, China; yanhongyan@hufe.edu.cn (H.Y.); 202005130214@mails.hufe.edu.cn (Z.Z.);
    201805130128@mails.hufe.edu.cn (Y.T.)
2   School of Management, Hangzhou Dianzi University, Hangzhou 310018, China; huanghanjie@hdu.edu.cn
3   School of Economics and Management, Changsha University of Science and Technology,
    Changsha 410000, China; zhouxy@stu.csust.edu.cn
4   School of Civil Engineering, Central South University, Changsha 410075, China
*   Correspondence: huping@hufe.edu.cn

**Abstract:** In view of the sociality, complexity, and uncertainty of major engineering projects, social stability poses many problems for social contradictions and conflicts in the whole life cycle of the project. This study aimed to investigate the approach of the coupling evaluation method to analyze the coupling influence of social stability risk factors of major projects. First, the potential risk factors of internal and external social stability risk of major projects were abstracted based on literature research and case analysis, and a bow-tie model and a coupling evaluation index system were constructed. Then, a N-K model of social stability risk coupling evaluation of major projects was constructed based on complex network, and the probability and risk value of the coupling of different risk factors were calculated. The studies showed that the coupling ways of social stability risk factors of major projects influence the social stability risk. Multi-factor risk coupling will increase the probability of social stability risk of major projects. The study of this paper provides a theoretical basis for the social stability risk management decision-making of major projects and promotes the sustainable development of major projects.

**Keywords:** major projects; social stability risk; risk factors; risk coupling evaluation

## 1. Introduction

Major projects are the symbol of human civilization, represent the progress of science and technology of the times, and reflect the degree of economic and social development. In recent years, a number of major infrastructure facilities have been established, designed, constructed and operated in China, such as the Three Gorges Water Conservancy Hub Project, South-to-North Water Transfer Project, West-to-East Gas Transmission Project, Hong Kong-Zhuhai-Macao Bridge, high-speed railroad network, etc. However, due to some special attributes of major engineering projects, such as the social nature, complexity, large scale, and uncertainty, the social risks caused during the construction process will seriously threaten the regional social stability. General Secretary Xi Jinping stressed at the opening ceremony of the seminar on major risks in 2019: improve the ability to prevent major risks and make efforts to resolve them so as to maintain sustainable and healthy economic development and social stability. Therefore, standing at the historical intersection of the "two hundred years" goal, in the macro context of China's social transformation and the vigorous construction of major projects, the construction of major projects should not only ensure a high-quality economic development, but also minimize the risks in order to maintain social stability.

At present, major engineering risks have been widely concerned and studied by the academic community, and a number of research results have been obtained. They mainly

focus on the social stability risk factors of major projects (Xiang P C, et.al., 2018 [1]; Munier N, et.al., 2016 [2]; Zhang R, et.al., 2016 [3]; Arukala S R, et.al., 2015 [4]), social stability risk evaluation (Lou X H, et.al., 2018 [5]; Li M, et.al., 2019 [6]; Khameneh A H, et.al., 2016 [7]; Zhou H, et.al., 2015 [8]; Yang SL, et.al., 2014 [9]), social stability risk governance (Zhang W, et.al., 2019 [10]; Cui C, et.al., 2012 [11]; Tan S, et.al., 2015 [12]; Huang Y J, et.al., 2013 [13]) and other aspects. Xiang Pengcheng et.al. [1] proposed that the key risk factors of social instability mainly include poor public opinion expression channels, weak government supervision and illegal project approval procedures. Luo Xiaohui et.al. [5] conducted a social stability risk assessment of major engineering projects under the two situational modes of black box operation and information disclosure for the four stages dynamic game model, and analyzed the impact of the feedback correction mechanism of social stability risk based on the hierarchical Bayesian network model, and they proposed that there were differences in the social stability risk assessment results of major engineering projects under different situations. Zhang Wei et.al. [10] identified 6 categories and 28 factors of social stability risk of major engineering projects, calculated the comprehensive driving force and comprehensive dependence of each risk based on Fuzzy ISM model, and put forward governance priority and governance measures for various risks. The above studies have promoted the development of social stability risk assessment theory for major projects, and provide a rich theoretical framework and knowledge reserve for this paper; however, the above studies all discussed the impact and evaluation of a single risk event on the social stability risk of major projects, ignoring the joint effect of multiple risk factors. The social stability risk of major projects has many influencing factors, the risk factors are strongly correlated, and the occurrence of accidents is often caused by the coupling of multiple factors. These characteristics are in line with the risk coupling analysis theory.

The N-K model [14,15] originated from information theory and is mainly used to analyze the influence of the interaction between the internal elements of the system on the overall adaptability of the system. It is widely used in economic and financial fields, and there is still a lack of research in the field of social stability risks in major projects. Therefore, this paper constructs the social stability risk coupling evaluation model based on the N-K model, and provides a reference for social stability risk management of major projects.

## 2. Identification of Risk Factors for Social Stability of Major Projects

The identification of risk factors for the social stability of major projects is a prerequisite for risk assessment and governance, and a necessary step before risk analysis and measures are taken. Social stability risk assessment of major projects will face many semi-structural decision-making problems, such as the lack of decision-making information, a large amount of false information, or excess information. However, comprehensively considering the three major factors of economy, society, and natural environment, the principle of triple bottom line (TBL) provides a new idea for the identification of risk factors and a new value standard for the sustainable development of an organization or society. For this reason, through the cases of major projects and the sudden process of group incidents, this paper analyzes the uncertain factors induced by the four-stage game process between both sides of the internal game (the government and the surrounding public), as well as external environment, including economic, natural, social and other exogenous uncertainties. Based on the summary of social stability risks and potential results of major projects, the Bow-tie model is constructed according to the logical sequence of "risk factor analysis-consequence evaluation-model creation".

### 2.1. Uncertainty Factors within the Main Players of the Game

In recent years, the social stability risks of major projects have been on the rise [16], especially in the relevant aspects of emergency decision-making that caused mass incidents. Therefore, on the basis of the principle of protecting the vital interests of the people to the greatest extent possible, it is necessary to resolve the value conflicts of different interest groups, such as the government and the public [17]. Major project construction is a sys-

tematic work integrating multi-field management and cooperation. Different stakeholders often face great conflicts of interest due to information asymmetry, benefit unbalance, and relationship cognitive dissonance, which seriously affects the sustainable development of major projects and triggers social stability risks. The main purpose of the game between the government and the public is to maximize their own interests. In this paper, the dynamic game process led by the local government and the public is divided into four stages (as shown in Figure 1 in order to explore the uncertain factors within the main game players of the major project [18].

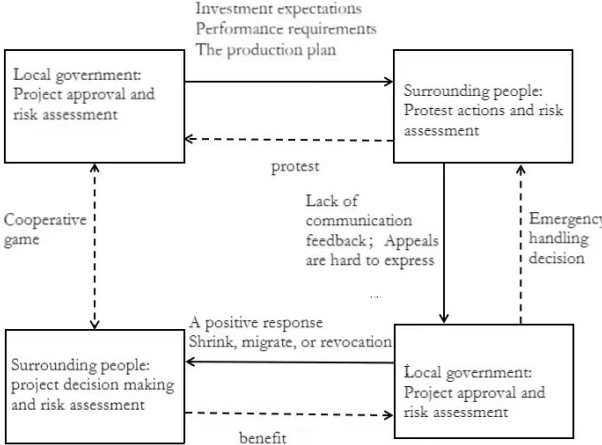

**Figure 1.** Dynamic game within the main players of the game. (The solid line represents the contradiction generation mechanism, and the dotted line represents the feedback correction system).

In the first stage, as major projects can promote local economic development and meet market demand, the government takes the first action and takes major projects as a decision-making node to realize investment expectations, performance pursuit, production planning, and so on. However, the public in the surrounding areas is worried that chemical products related to major projects will threaten their own safety and health, the quality of the surrounding environment, and the asset chain, and this is taken as the starting point of the game. In the second stage, the surrounding public takes protest actions as the decision-making node according to the first decision-making action of the local government and as a post-actor. When there is no reasonable channel to express their own demands or when they are unable to obtain reasonable and legitimate interest demands in all aspects of major projects, the surrounding public has to stimulate the high spirits of the masses and set off social protests. If the local government actively pays attention to the interests of the public and can communicate with the public about project risks as soon as possible, the public can understand the risk level and prevention and control measures of the project, which will help the public to accept the risk assessment conclusions of the project and form an objective perception of risks, and thus the public may abandon the protests and embrace the project [19]. In the third stage, the government must make scientific and reasonable emergency response decisions, and some governments build an effective interaction mechanism with the public to dispel public doubts; while some local governments take decisions such as reducing, relocating, or even canceling major projects to calm the social situation. In the fourth stage, the surrounding public decides whether to end the protest action so as to achieve the cooperative game according to the emergency response decision of the local government.

### 2.2. External Environmental Factors

Through the analysis of major engineering accident cases, it is found that there are endless cases of mass incidents caused by external environmental factors such as compensation policies, safety accidents, and environmental pollution. Therefore, external environmental factors are also an important part of the risks that threaten the social stability of major

projects. In 1997, John Elkington put forward the representative "triple-bottom-line" theory of social responsibility. He believed that the foundation for an organization to achieve sustainable development is to seek the balance of economic, social, and environmental responsibility on the basis of bottom-line responsibility. Therefore, this paper analyzes the external uncertain factors of the social stability risk of major projects based on the triple bottom line principle of "economy-society-natural environment" (as shown in Figure 2).

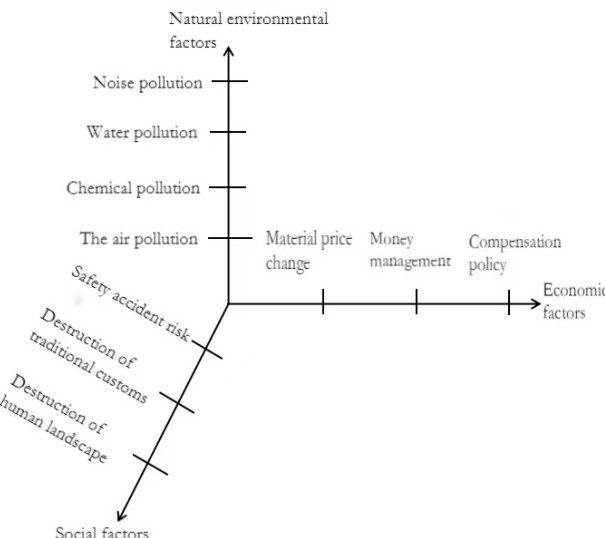

**Figure 2.** External uncertain factors.

Among the external uncertainties, the first is economic factors [1,16]. The construction period of major projects is long, there are many relevant interest groups, and the construction technology is complex, therefore, there are many unmeasurable risks, and it is easy to be affected by economic factors, including a change in material price, labor supply, fund management, and compensation policy.

The second is the social factors [1,16]. Based on previous literature review, this paper believes that the social factors affecting major projects mainly include the following two aspects: on the one hand, it is the risk of safety accidents brought about by major projects, such as the "five major injuries" of common safety risks; on the other hand, it is the risk caused by the destruction of the surrounding cultural landscape and customs.

Finally, there are natural environmental factors [1,16], through on-the-spot investigation and case analysis, the waste of resources caused by land development, water pollution, chemical pollution, air pollution, waste pollution, and so on, are the triggers of surrounding public protests, parades, and even mass disturbances.

### 2.3. Building a Bow-Tie Model Based on Internal and External Factors Analysis

The Bow-tie model is a risk management tool that organically combines fault tree analysis and event tree analysis [20]. It is a risk analysis method with strong operability and high visualization. By drawing the bow tie diagram, the potential risk factors of the accident are put on the left as the fault tree part, and the results caused by the accident are put on the right as the event tree part, and lists threats and barriers to reflect the logical development of the event, then build a graphical model. Based on the above analysis, the internal risk factors of social stability of major projects are mainly reflected in the policy risk and public risk induced by the game process between the government and the surrounding public, and the external risk factors are mainly reflected in the economic risk, natural risk and social risk generated by the external environment. Therefore, based on the traditional bow-tie model and the logical idea of "risk source-consequence-evaluation barrier setting", a series of indicators related to the social stability risk of major projects were determined through the forward and reverse push of risk source and consequence, Furthermore, the

risk countermeasures of practical value were put forward, and the bow-tie model was constructed (as shown in Figure 3).

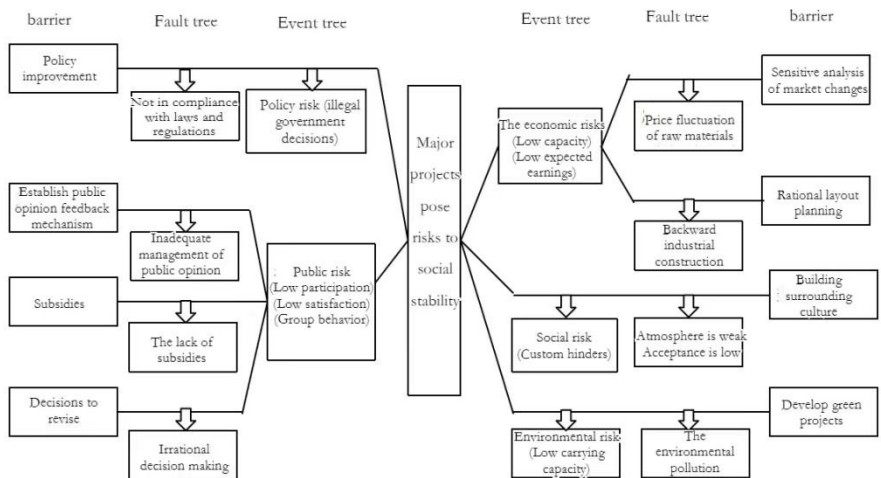

**Figure 3.** Bow-tie model of social stability risk factors for major projects.

## 3. Evaluation of Social Stability Risk Coupling for Major Projects

### 3.1. Construction of Risk Evaluation Index System for Social Stability for Major Projects

Based on the identification of social stability risk factors of major projects and the Bow-tie model, this paper constructs the dimensions of the social stability risk evaluation index system of major projects, including government risk, public risk, economic risk, social risk, and natural environmental risk; comprehensively analyzes the internal uncertain factors of the local government and the surrounding residents, as well as the exogenous uncertain factors such as "economic-social-natural environment" produced by the external environment, dynamically and dialectically considers the risk factors, and summarizes them; and selects 12 program layer indicators according to the principles of comprehensiveness, science, maneuverability, and humanization, as shown in Table 1.

**Table 1.** Indicators at the program level.

| Criterion Layer | Scheme Layer | Indicator Description |
|---|---|---|
| Government risk | Decision legitimacy | The government abides by laws and regulations, technical standards, and contract norms in decision-making |
| | Affiliation institutional perfection | Establish and implement relevant systems at all stages to ensure the smooth implementation of the project and the interests of the relevant masses |
| Public risk | Social participation | The participation of the public in putting forward reasonable suggestions in major engineering fields |
| | Social satisfaction | Satisfaction of the public to enjoy environmental subsidies and other preferential policies |
| | Risk of group behavior | Group events such as group strikes, demonstrations, disturbances, petitions by the masses, etc. |
| Economic risk | Price changes | The price fluctuation caused by raw materials or supply and demand has an impact on the original profit structure |
| | Mode of operational management | Failure to coordinate the various management elements, resulting in more labor and material consumption |
| Social risk | Accident safety risks | Building collapse, occurrence of fire, explosion accidents, etc. |
| | Traditional customs influence | Migrant masses need to accept the customs of the placement area, culture relearning, etc. |

**Table 1.** *Cont.*

| Criterion Layer | Scheme Layer | Indicator Description |
|---|---|---|
| Natural environmental risk | Air pollution | When the long-term emission of toxic and harmful gases reaches a certain degree of pollution, it will cause irreversible damage to the atmosphere |
| | Water pollution | Pollution of water quality caused by the discharge of toxic substances and waste water during construction and production |
| | Resource occupation | Occupation of surrounding resources by major projects |

*3.2. Coupling Evaluation of Social Stability Risks of Major Projects Based on N-K Model*

The N-K model [14,15] is a general model used to study complex dynamic systems, including two parameters: N is the number of constituent elements of the system, and K is the number of interdependencies in the network. If there are N elements in the system, and each element has n different states, then there are N kinds of possible combinations. The elements of the system are combined in a certain way, that is, a network is formed. The minimum value of K is 0 and the maximum value is N-1.

The steps of using the N-K model to measure the social stability risk coupling of major projects include: major project coupling risk classification, data statistics, and coupling probability calculation.

According to the number of risk factor coupling, the social stability risk coupling of major projects is divided into the following three categories:

(1) Single factor coupling risk: A single risk factor affecting the social stability of major projects will contain multiple risk factors, and each risk factor will interact with each other. Single factor coupling risk includes government (abbreviated G, Code a) factor risk, public (abbreviated P, Code b) factor risk, economic (abbreviated E, Code c) factor risk, social (abbreviated S, Code d) factor risk and natural environmental (abbreviated NE, Code e) factor risk, are recorded as $T_{10}$ (*a*), $T_{11}$ (*b*), $T_{12}$ (*c*), $T_{13}$ (*d*), $T_{14}$ (*e*), respectively, and the total value of coupling risk is recorded as $T_1$. The single factor coupling risk is shown in Table 2.

**Table 2.** Single factor coupling risk.

| Type | Government Factor Risk | Public Risk | Economic Risk | Social Risk | Natural Environmental Risk |
|---|---|---|---|---|---|
| Expression | $T_{10}$ (*a*) | $T_{11}$ (*b*) | $T_{12}$ (*c*) | $T_{13}$ (*d*) | $T_{14}$ (*e*) |

(2) Two-factor coupling risk: Includes 10 types of two-factor coupling risk, and the total value of coupling risk is recorded as $T_2$. The two-factor coupling risk is shown in Table 3.

**Table 3.** Two-factor coupling risk.

| Type | G-P Coupling Risk | G-E Coupling Risk | G-S Coupling Risk | G-NE Coupling Risk | P-E Coupling Risk |
|---|---|---|---|---|---|
| Expression | $T_{20}$ (*a,b*) | $T_{21}$ (*a,c*) | $T_{22}$ (*a,d*) | $T_{23}$ (*a,e*) | $T_{24}$ (*b,c*) |
| Type | P-S coupling risk | P-NE coupling risk | E-S coupling risk | E-NE coupling risk | S-NE coupling risk |
| Expression | $T_{25}$ (*b,d*) | $T_{26}$ (*b,e*) | $T_{27}$ (*c,d*) | $T_{28}$ (*c,e*) | $T_{29}$ (*d,e*) |

(3) Multi-factor coupling risk: Refers to the interaction of three or more risk factors affecting the social stability of major projects, and the total value of coupling risk is recorded as $T_3$. The multi-factor coupling risk is shown in Table 4.

**Table 4.** Multi-factor coupling risk.

| Type | G-P-E Coupling Risk | G-P-S Coupling Risk | G-P-NE Coupling Risk | G-E-S Coupling Risk | G-E-NE Coupling Risk | G-S-NE Coupling Risk |
|------|------|------|------|------|------|------|
| Expression | $T_{30}$ (a,b,c) | $T_{31}$ (a,b,d) | $T_{32}$ (a,b,e) | $T_{33}$ (a,c,d) | $T_{34}$ (a,c,e) | $T_{35}$ (a,d,e) |
| Type | P-E-S coupling risk | P-E-NE coupling risk | P-S-NE coupling risk | E-S-NE coupling risk | G-E-S-NE coupling risk | G-P-S-NE coupling risk |
| Expression | $T_{36}$ (b,c,d) | $T_{37}$ (b,c,e) | $T_{38}$ (b,d,e) | $T_{39}$ (c,d,e) | $T_{310}$ (a,c,d,e) | $T_{311}$ (a,b,d,e) |
| Type | G-P-E-NE coupling risk | G-P-E-S coupling risk | P-E-S-NE coupling risk | G-P-E-S-NE coupling risk | - | - |
| Expression | $T_{312}$ (a,b,c,e) | $T_{313}$ (a,b,c,d) | $T_{314}$ (b,c,d,e) | $T_4$ (a,b,c,d,e) | - | - |

In this paper, by calculating the interactive information among five types of social stability risk factors of major projects, the coupling effect is evaluated to form a new risk state. The probability that this type of coupling occurs is measured in terms of the number of times that it occurs more rapidly. The coupling risk magnitude and the accident probability are measured in terms of the coupling value magnitude, i.e., if the resulting value healed with some form of coupling, then the coupling risk healed with the resulting probability healed.

Firstly, the calculation formula of single factor coupling is shown in Formula (1).

$$T(a,b,c,d,e) = \sum_{h=1}^{H} \sum_{i=1}^{I} \sum_{j=1}^{J} \sum_{k=1}^{K} \sum_{l=1}^{L} [P_{hijkl} \times \log_2(\frac{P_{hijkl}}{P_{h....} \times P_{.i...} \times P_{..j..} \times P_{...k.} \times P_{....l}})] \quad (1)$$

where $a$, $b$, $c$, $d$ and $e$ represent five coupling element numbers (where a represents government risk, b represents public risk, c represents economic risk, d represents social risk, and e represents natural environmental insurance); $T$ represents the coupling value, and the larger the coupling value is, the more likely the risk accident caused by this method is; $h$, $i$, $j$, $k$, $l$ represent the state of the five factors respectively; $P_{hijkl}$ represents the probability of the coupling of the five factors; $P_{h....}$ represents when the government risk factor is in the h state, the single factor coupling probability; $P_{.i...}$ represents the probability of single factor coupling when the public risk factor is in the $i$ state; $P_{..j..}$ represents the probability of single factor coupling when the economic risk factor is in the $j$ state; $P_{...k.}$ represents the probability of single factor coupling when the social risk factor is in $k$ state; $P_{....l}$ represents the probability of single factor coupling when the natural environmental risk factor is in the $l$ state.

Two-factor coupling refers to a form of pairwise coupling among the risk coupling factors of social stability in major projects. The two risk couplings will produce 10 cases; taking $T_{20}$ ($a$, $b$) as an example, its calculation formula is shown in Formula (2).

$$T_{21}(a,b) = \sum_{h=1}^{H} \sum_{i=1}^{I} [P_{hi...} \times \log_2(\frac{P_{hi...}}{P_{h....} \times P_{.i...}})] \quad (2)$$

Multi-factor coupling refers to the interaction of more than two factors in the coupling factors that affect the social stability risk of major projects, with a total of 13 cases; taking $T_{30}$ ($a$, $b$, $c$) as an example, its calculation formula is shown in Formula (3).

$$T_{31}(a,b,c) = \sum_{h=1}^{H} \sum_{i=1}^{I} \sum_{j=1}^{J} [P_{hij..} \times \log_2(\frac{P_{hij..}}{P_{h....} \times P_{.i...} \times P_{..j..}})] \quad (3)$$

Finally, according to the order of each coupling value, the conclusion of coupling evaluation is drawn.

## 4. Example

### 4.1. Example Statistics of Social Stability Risk Events in Major Projects

This paper collected from website news reports, papers, paper press publications at home and abroad to analyze the cases of stable risk events of major engineering societies at home and abroad, and counted 108 risk events occurring at home and abroad between 2000 and 2020, including 72 risk events at home and 36 risk events abroad; the major engineering social stability risk events are shown in Table 5.

**Table 5.** Information on social stability risk events of major projects.

| Project Name | Government Risk Factors | Public Risk Factors | Economic Risk Factors | Social Risk Factors | Natural Environmental Risk Factors |
|---|---|---|---|---|---|
| Three Gorges Project (China) | No | Social satisfaction problem | No | Traditional customs problem | Resource occupation problem |
| Hong Kong-Zhuhai-Macao Bridge (China) | No | Social participation problem | The problem of the mode of management | No | No |
| Bird's Nest (China) | No | Social satisfaction problem | The problem of the mode of management | No | No |
| New Federal Building of San Francisco (United States) | Legitimacy of decision | Other | The problem of the mode of management | No | No |
| Kemper thermal power plants (United States) | No | No | The problem of the mode of management | No | Air Pollution |
| Sampoong Department Store(South Korea) | Legitimacy of decision | Social satisfaction problem | Other | Risk of safety accident | No |

In the model, 1 and 0 are used to indicate whether each factor is in an unsafe state, 1 indicates occurrence, and 0 indicates that the risk occurrence probability is not present and the coupling value is used to quantify the risk occurrence probability (as shown in Table 6). The coupling factors with a frequency (frequency) of 0 on the way are not marked.

**Table 6.** Statistics of social stability risk events of major projects under different coupling modes.

| Single Factor Coupling | | | Two-Factor Coupling | | | Multi-Factor Coupling | | |
|---|---|---|---|---|---|---|---|---|
| Coupling Factor | Frequency/ Time | Frequency | Coupling Factor | Frequency/ Time | Frequency | Coupling Factor | Frequency/ Time | Frequency |
| 10,000 | 3 | 0.028 | 11,000 | 7 | 0.065 | 11,010 | 4 | 0.037 |
| 01,000 | 9 | 0.083 | 10,100 | 2 | 0.019 | 10,110 | 1 | 0.009 |
| 00,100 | 5 | 0.046 | 10,010 | 4 | 0.037 | 10,011 | 2 | 0.019 |
| 00,010 | 22 | 0.204 | 01,100 | 1 | 0.009 | 01,011 | 6 | 0.056 |
| 00,001 | 6 | 0.055 | 01,010 | 30 | 0.278 | - | - | - |
| - | - | - | 01,001 | 5 | 0.046 | - | - | - |
| - | - | - | 00,011 | 1 | 0.009 | - | - | - |

In Table 3, 00000 of the single factor coupling indicated that none of the five coupling factors had an impact on the social stability risk of major projects, 10,000 said that only government factors had an impact on the social stability risk of major projects, and there were three such accidents with a frequency of 0.028; 11,000 in the two factor coupling indicates that the risk is the result of the coupling of government factors as well as social public factors, and there are seven such accidents with a frequency of 0.065; 11,010 of the

multi-factor coupling indicates that the risk is the result of the coupling of government factors, social public factors, and social factors, and there are four such accidents with a frequency of 0.037.

### 4.2. Risk Coupling Value Calculation

(1)　Single factor coupling probability

By analyzing the statistical data of social stability risk accidents of major projects, the probability that government factors do not affect the social stability risk of major projects is as follows: $P_{0\ldots.} = P_{01,000} + P_{00,100} + P_{00,010} + P_{0,0001} + P_{01,100} + P_{01,010} + P_{01,001} + P_{00,110} + P_{00,101} + P_{00,011} + P_{01,110} + P_{01,011} + P_{00,111} + P_{0,1111} = 0.786$. Similarly, $P_{1\ldots.}$, $P_{.0\ldots}$, $P_{.1\ldots}$, $P_{..0..}$, $P_{..1..}$, $P_{\ldots0.}$, $P_{\ldots1.}$, $P_{\ldots.0}$, $P_{\ldots.1}$ can be calculated, and the calculated results were tabulated in Table 7.

**Table 7.** Single factor coupling probability.

| Coupling Mode | Probability | Coupling Mode | Probability |
|---|---|---|---|
| $P_{0\ldots.}$ | 0.786 | $P_{..1..}$ | 0.148 |
| $P_{1\ldots.}$ | 0.214 | $P_{\ldots0.}$ | 0.351 |
| $P_{.0\ldots}$ | 0.426 | $P_{\ldots1.}$ | 0.649 |
| $P_{.1\ldots}$ | 0.574 | $P_{\ldots.0}$ | 0.815 |
| $P_{..0..}$ | 0.852 | $P_{\ldots.1}$ | 0.185 |

(2)　Tow-factor coupling probability

By analyzing the statistical data of social stability risk accidents of major projects, the probability of accidents when government and public factors do not participate in risk coupling is $P_{00\ldots} = P_{00,000} + P_{00,100} + P_{00,010} + P_{00,001} + P_{00,110} + P_{00,101} + P_{0,0011} + P_{00,111} = 0.314$. Similarly, $P_{01\ldots}$, $P_{10\ldots}$, $P_{11\ldots}$ can be calculated, and the calculated results were shown in Table 8.

**Table 8.** Two-factor coupling probability.

| Coupling Mode | Probability | Coupling Mode | Probability | Coupling Mode | Probability | Coupling Mode | Probability |
|---|---|---|---|---|---|---|---|
| $P_{00\ldots}$ | 0.314 | $P_{1..0.}$ | 0.112 | $P_{.0.0.}$ | 0.148 | $P_{..10.}$ | 0.074 |
| $P_{01\ldots}$ | 0.472 | $P_{1..1.}$ | 0.102 | $P_{.0.1.}$ | 0.278 | $P_{..11.}$ | 0.009 |
| $P_{10\ldots}$ | 0.103 | $P_{0\ldots0}$ | 0.287 | $P_{.1.0.}$ | 0.203 | $P_{..0.0}$ | 0.741 |
| $P_{11\ldots}$ | 0.102 | $P_{0\ldots1}$ | 0.166 | $P_{.1.1.}$ | 0.371 | $P_{..0.1}$ | 0.185 |
| $P_{0.0..}$ | 0.731 | $P_{1\ldots0}$ | 0.195 | $P_{.0..0}$ | 0.343 | $P_{..1.0}$ | 0.083 |
| $P_{0.1..}$ | 0.055 | $P_{1\ldots1}$ | 0.019 | $P_{.0..1}$ | 0.083 | $P_{..1.1}$ | 0 |
| $P_{1.0..}$ | 0.186 | $P_{.00..}$ | 0.352 | $P_{.1..0}$ | 0.472 | $P_{\ldots00}$ | 0.25 |
| $P_{1.1..}$ | 0.028 | $P_{.01..}$ | 0.074 | $P_{.1..2}$ | 0.102 | $P_{\ldots01}$ | 0.101 |
| $P_{0.0..}$ | 0.184 | $P_{.10..}$ | 0.565 | $P_{..00.}$ | 0.277 | $P_{\ldots10}$ | 0.565 |
| $P_{0.1..}$ | 0.547 | $P_{.11..}$ | 0.009 | $P_{..01.}$ | 0.64 | $P_{\ldots11}$ | 0.084 |

(3)　Multi-factor coupling probability

By analyzing the statistical data of social stability risk accidents of major projects, the probability of accidents when the government, the public, and economic factors do not participate in the risk coupling is $P_{000..} = P_{00,000} + P_{00,001} + P_{0,0010} + P_{00,011} = 0.268$. Similarly, $P_{100..}$, $P_{010..}$, $P_{001..}$ can be calculated, in which Tables 9 and 10 present the results.

**Table 9.** Three-factor coupling probability.

| Coupling Mode | Probability | Coupling Mode | Probability | Coupling Mode | Probability | Coupling Mode | Probability |
|---|---|---|---|---|---|---|---|
| $P_{000..}$ | 0.268 | $P_{1.01.}$ | 0.056 | $P_{.100.}$ | 0.148 | $P_{.1.01}$ | 0.046 |
| $P_{100..}$ | 0.084 | $P_{1.11.}$ | 0.009 | $P_{.001.}$ | 0.269 | $P_{.0.11}$ | 0.019 |
| $P_{001..}$ | 0.046 | $P_{0..00}$ | 0.138 | $P_{.110.}$ | 0.009 | $P_{.1.11}$ | 0.056 |
| $P_{110..}$ | 0.102 | $P_{1..00}$ | 0.112 | $P_{.101.}$ | 0.371 | $P_{..000}$ | 0.176 |
| $P_{101..}$ | 0.028 | $P_{0..01}$ | 0.101 | $P_{.011.}$ | 0.009 | $P_{..100}$ | 0.074 |
| $P_{011..}$ | 0.009 | $P_{1..10}$ | 0.074 | $P_{.0.00}$ | 0.093 | $P_{..001}$ | 0.101 |
| $P_{0.00.}$ | 0.184 | $P_{0..11}$ | 0.009 | $P_{.1.00}$ | 0.157 | $P_{..110}$ | 0.009 |
| $P_{1.00.}$ | 0.093 | $P_{1..11}$ | 0.019 | $P_{.0.01}$ | 0.055 | $P_{..011}$ | 0.084 |
| $P_{0.01.}$ | 0.547 | $P_{.000.}$ | 0.083 | $P_{.1.10}$ | 0.315 | $P_{1.10.}$ | 0.019 |

**Table 10.** Four-factor coupling probability.

| Coupling Mode | Probability | Coupling Mode | Probability | Coupling Mode | Probability | Coupling Mode | Probability |
|---|---|---|---|---|---|---|---|
| $P_{0000.}$ | 0.055 | $P_{0.000}$ | 0.083 | $P_{10.00}$ | 0.047 | $P_{01.11}$ | 0.056 |
| $P_{1000.}$ | 0.028 | $P_{1.000}$ | 0.093 | $P_{01.00}$ | 0.092 | $P_{000.0}$ | 0.204 |
| $P_{0100.}$ | 0.129 | $P_{0.100}$ | 0.055 | $P_{00.10}$ | 0.204 | $P_{100.0}$ | 0.065 |
| $P_{0010.}$ | 0.046 | $P_{0.010}$ | 0.482 | $P_{00.01}$ | 0.055 | $P_{010.0}$ | 0.361 |
| $P_{0001.}$ | 0.213 | $P_{0.001}$ | 0.101 | $P_{11.00}$ | 0.065 | $P_{001.0}$ | 0.046 |
| $P_{1100.}$ | 0.065 | $P_{1.100}$ | 0.019 | $P_{10.10}$ | 0.046 | $P_{000.1}$ | 0.064 |
| $P_{1010.}$ | 0.019 | $P_{1.010}$ | 0.074 | $P_{01.10}$ | 0.278 | $P_{110.0}$ | 0.102 |
| $P_{1001.}$ | 0.056 | $P_{0.011}$ | 0.065 | $P_{01.01}$ | 0.046 | $P_{101.0}$ | 0.028 |
| $P_{0110.}$ | 0.009 | $P_{1.110}$ | 0.009 | $P_{00.11}$ | 0.009 | $P_{100.1}$ | 0.019 |
| $P_{0101.}$ | 0.278 | $P_{1.011}$ | 0.019 | $P_{11.10}$ | 0.037 | $P_{011.0}$ | 0.009 |

(4) T value calculation

The T value can be obtained according to the Formulas (1)–(3) as shown in Table 11.

**Table 11.** T values under different coupling regimes.

| Coupling Mode | T Values | Coupling Mode | T Values | Coupling Mode | T Values |
|---|---|---|---|---|---|
| $T_{20}$ | 0.084 | $T_{29}$ | 0.046 | $T_{39}$ | 0.140 |
| $T_{21}$ | 0.033 | $T_{31}$ | 0.248 | $T_{310}$ | 0.252 |
| $T_{22}$ | 0.101 | $T_{32}$ | 0.141 | $T_{311}$ | 0.320 |
| $T_{23}$ | −0.001 | $T_{33}$ | 0.188 | $T_{312}$ | 0.196 |
| $T_{24}$ | 0.076 | $T_{34}$ | 0.099 | $T_{313}$ | 0.408 |
| $T_{25}$ | 0.179 | $T_{35}$ | 0.173 | $T_{314}$ | 0.304 |
| $T_{26}$ | 0.034 | $T_{36}$ | 0.202 | $T_4$ | 0.579 |
| $T_{27}$ | 0.099 | $T_{37}$ | 0.138 | | |
| $T_{28}$ | −0.003 | $T_{38}$ | 0.197 | | |

*4.3. Conclusion and Discussion of Example Risk Coupling Evaluation*

The following conclusions can be drawn from the present study:

(1) The more kinds of coupling risk factors, the greater the risk of social stability of major projects is. From the calculation results, it can be inferred that the five-factor coupling risk value ($T_4 = 0.579$) is greater than the four-factor coupling risk value ($T_{310}$–$T_{314}$), the four-factor coupling risk value ($T_{310}$–$T_{314}$) is generally greater than the three-factor coupling risk value ($T_{30}$–$T_{39}$), and the three-factor coupling risk value ($T_{30}$–$T_{39}$) is generally greater than the two-factor coupling risk value ($T_{20}$–$T_{29}$), which is consistent with the actual situation of social stability risks of major projects.

(2) Among the four-factor coupling risks, the coupling value of the government-public-economic-social factor ($T_{313} = 0.408$) is the largest, and that of the government-public-economic-natural environmental factor is the smallest ($T_{312} = 0.196$). At the same time, the coupling value of the government-public-social-natural environmental factor ($T_{311} = 0.320$) is larger than that of the government-economic-social-natural environmental factor ($T_{310} = 0.252$), and it is between the coupling value of the government-public-economic-social factor and the government-public-economic-natural environmental factor. Among the three-factor coupling risks, the coupling value of government-public-social factors ($T_{31} = 0.248$) is the largest, while that of government-economic-natural environmental factors ($T_{34} = 0.099$) is the smallest, which shows that social factors and social public factors play a greater role in major project risks, and the range of social factors is relatively wide, which can affect other factors to a certain extent. Among the social factors, safety accidents not only pose a threat to people's lives and property, but also cause huge economic losses directly or indirectly to society, thus affecting social stability; the public will take the hidden dangers of accidents, policy subsidies, environmental pollution as a fuse to cause social stability risks.

(3) Among the two-factor coupling risks, the coupling value of government-economic ($T_{21} = 0.033$) < public-natural environmental ($T_{26} = 0.034$) < social-natural environmental ($T_{29} = 0.046$) < public-economic ($T_{24} = 0.076$) < government-public ($T_{20} = 0.084$) <economic-social ($T_{27} = 0.099$) < government-social ($T_{22} = 0.101$) < public-social ($T_{25} = 0.179$), therefore the value of public-social coupling risk is the largest median risk of two factor coupling, and there is a great coupling between social factors and social public factors. Among the social factors, the destruction of traditional customs caused by major projects has a far-reaching impact on the public, that is, the integration of land expropriation immigrants and residents in resettlement areas. For example, the "Three Gorges Project" involves the migration of nearly two million people. These people may face the risk of losing land, declining living standards, unemployment, marginalization, and broken community relations due to a lack of a sense of security and sense of belonging after resettlement. In addition, immigrants who leave their homes not only need to learn different languages and cultures, but also accept local customs, all of which are social factors that may endanger the stability of the local society.

(4) From the multi-factor and two-factor coupling risk, we can see that the coupling values of politics-society ($T_{22} = 0.101$) and economy-society ($T_{27} = 0.099$) are relatively large and similar. Therefore, among the government factors, the legitimacy, rationality, and information transparency of government policy, the change of raw material prices, and the coupling between capital chain management and social risk factors are relatively strong.

*4.4. Coupling Risk Countermeasure*

(1) In order to solve the risk of social-public coupling, one of the meeting points is the media. On the one hand, the public should gradually cultivate the awareness of finding the media for something. When the risks of major projects infringe upon the vital interests of the public, the public should exercise their power within the scope of the law to seek help from the media or pretend to be the media themselves. Through the official channel the network platform can be used to output information to attract attention, improve the ability of thinking and their own comprehensive quality, and strengthen the ability to screen information. On the other hand, the media should strengthen the networking and digitization of information feedback and define the social responsibility objectives of major projects. Through press conferences, Weibo interviews, large forums, and other information media, the integration of public subsidies, environmental feedback, and other data to analyze the causes of the risk of social stability, to answer questions to the public to form a good interaction.

(2) When making decisions on major projects, the government should, on the premise of abiding by national laws and regulations, technical norms, and industry standards, focus on the disclosure of relevant information in the field of approval and implementation of major projects in an all-round way, and show the information to the society and the public in an open and transparent manner. However, the social public group behavior risk is mostly caused by unreasonable decision-making, therefore it is necessary to construct the concept of overall governance and pursue benign interaction for decision-making revision under the guidance of a people-oriented concept.

(3) An early risk warning mechanism for social stability will be formed. The social stability risk of major projects increases with the increase of risk coupling factors, thus it is necessary to predict and warn of the risk factors before the occurrence of social stability risk events. On the one hand, more serious multi-factor coupling risk events can be avoided through the early warning mechanism; on the other hand, the abnormal indicators in the early warning mechanism can be traced back to the hidden risk factors to obtain more efficient and accurate social stability risk management programs and countermeasures.

## 5. Conclusions

The complexity of major projects makes them prone to have various conflicts of interest in the construction process, leading to the occurrence of group events and the risk of social stability. Based on the analysis of the risk events related to the social stability risk of major projects, the authors constructed a social stability risk coupling evaluation model to study the impact of the coupling of different factors on the social stability risk of major projects.

The main conclusions of this paper are as follows: (1) Five factors, such as government, social public, economy, society, and natural environment, are the main risk factors that affect the social stability of major projects. (2) There are many ways of coupling among the five factors, and the risk of social stability caused by different coupling modes is different, among the five factors coupling, the social stability risk caused by the coupling of the five factors of "government-public-society-environmental-economy" is the greatest risk. In addition, the "politics-public-economy-society" coupling mode among the four factors coupling, the "politics-public-society" coupling mode among the three factors, and the "public-society" coupling mode among the two factors are most at risk. (3) Overall, there is a positive correlation between coupling factors and risk probability, that is: five-factor coupling risk > four-factor coupling risk > three-factor coupling risk > two-factor coupling risk. From a local perspective, some three-factor coupling risks are higher than four-factor coupling risks, such as "government-public-social" coupling risk > "government-public-economic-environmental" coupling risk. (4) Based on the conclusion of social stability risk coupling evaluation of major projects, some countermeasures were put forward.

The NK analysis method is an extension of traditional analysis methods. Traditional analysis approaches do not consider the interdependence between various risks, such as the analytic hierarchy process [21] and the fuzzy method [22]. It is difficult to quantitatively evaluate the coupling relationship between various risk factors. In this study, the NK model can be used to analyze the risk value of the social stability risk of major projects coupled with different risk factors quantitatively, therefore some coupling laws of social stability risk of major projects were drawn. The research of this paper provides a method for decision-makers to assess the social stability risk of major projects, and provides a theoretical basis for the decision-making of social stability risk management of major projects.

There are some requirements on the integrity of the data and the number of samples for the model that were constructed in this paper. Therefore, in the follow-up research, the integrity of the data can be continuously improved, the sample size can be expanded, and the calculation accuracy can be improved to make the calculation results more in line with the actual situation.

**Author Contributions:** Conceptualization, H.Y. and P.H.; methodology, Y.T.; validation, Z.Z., Y.T.; formal analysis, H.Y. and X.Z.; investigat.ion, X.Z., Z.Z. and H.H.; data curation, Y.T.; writing—original

draft preparation, H.Y.; writing—review and editing, P.H.; supervision, P.H.; project administration, H.Y.; funding acquisition, H.Y. and P.H. All authors have read and agreed to the published version of the manuscript.

**Funding:** This research was funded by [the Natural Science Foundation of Hunan Province] grant number [2021JJ30748], [Scientific Research Key Project of Hunan Provincial Department of Education] grant number [21A0207], [National University Student Innovation and Entrepreneurship Project] grant number [S202011532017], [National University Student Innovation and Entrepreneurship Project] grant number [S202111532023], [Hunan University student innovation and Entrepreneurship Project] grant number [4059].

**Conflicts of Interest:** The authors declare no conflict of interest.

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
