# Peer review of "Risk Coupling Evaluation of Social Stability of Major Engineering Based on N-K Model"

_buildings, doi:10.3390/buildings12060702_

Round 1

Reviewer 1 Report

The paper describes an interesting topic, suitable for this journal. My comments as follows: 

ABSTRACT

-The authors need to improve English writing style as their sentences are very confusing. Please rewrite. 

INTRODUCTION

  • The authors should explain in a better way the extensive list of references provided in this section.

IDENTIFICATION OF RISK FACTOR FOR SOCIAL STABILITY

  • Please provide references across this section. I just counted 1 reference in this section. There are many instances in which authors need to provide reference to indicate they are using ideas proposed by other researchers.
  • Please explain Figure 2 in a better way. 
  • Is it Figure 3 the result of this research or is it something obtained from the literature? If it is the result of this research, how did the authors get to this result? If it is taken from literature, please provide references. 

EVALUATION OF SOCIAL STABILITY RISK COUPLING FOR MAJOR PROJECTS

  • Please provide references across this section. 
  • Please consider to include a Table for describing all the explain factors and risks as they way authors record these indicators is confusing. For instance, I am not sure what T10(a) means. 

EXAMPLE

  • Please discuss the validity of your results

CONCLUSIONS

  • Please incorporate theoretical contributions and practical implications. 

Author Response

Dear  Reviewer:

Thank you for your letter and for the reviewers’ comments concerning our manuscript entitled “Risk Coupling Evaluation of Social Stability of Major Engineering Based on N-K Model”. Those comments are all valuable and very helpful for revising and improving our paper, as well as the important guiding significance to our researches. We have studied comments carefully and have made correction which we hope meet with approval. Revised portion are marked in red in the paper. The main corrections in the paper and the responds to the reviewer’s comments are attached.

Reviewer 2 Report

This paper is actually quite interesting and written in a good writing flow. However, I still have important issues that need to be addressed by the authors. I offer my detailed, section-wise comments below:

  1. It does not add significant value to the existing body of literature. Furthermore, there is no significant research conducted in related areas and only a few research papers seem to have been included in this work. The authors need to include references for explaining the method and the other techniques in the literature review for social stability risk assessment. They need to justify based on the literature review why they selected those “external uncertain factors”, why they defined that bow-tie model, why they chose the indicators at the program level. They do not use any reference along the sections 2, to identify the risk factors, and section 3, to explain the method.
  2. Knowledge gap. Lines 45 to 50. The authors focus on showing the most important papers about social stability risk factors of major projects. Still, they really need to explain why these papers are important for their research. The knowledge gap is lacking in the document.
  3. Discussion. The authors do not discuss the results based on the literature.

Author Response

Dear  Reviewer:

Thank you for your letter and for the reviewers’ comments concerning our manuscript entitled “Risk Coupling Evaluation of Social Stability of Major Engineering Based on N-K Model”. Those comments are all valuable and very helpful for revising and improving our paper, as well as the important guiding significance to our researches. We have studied comments carefully and have made correction which we hope meet with approval. Revised portion are marked in red in the paper. The main corrections in the paper and the responds to the reviewer’s comments are as attached.

Reviewer 3 Report

The main purpose of this research is to develop a social stability risk model based on the N-K model that can be used to manage social stability risk in major projects. Such study is essential for the scientific community as well as the construction company, local government, and public. Overall, the description is accurate and the ideas are well organized. However, the research shows several limitations that are explained below.

  1. Page 2, Line 88: Figure 1 depicts the dynamic game process led by the local government and the public. Public awareness policy should be discussed in Fig. 1. This policy may help the public in appreciating the significance of major projects.

  1. Page 4, Line 133: ‘The main economic factors are the change of material price, fund management, and compensation policy’. As the construction period of major projects is long, therefore, labour crisis and uncertainty may also affect the economic factors. This should be discussed in detail here.

  1. Page 4, Line 140: Waste pollution should be incorporated with other natural environmental factors. The waste management policies may reduce the uncertainty associated with natural environmental risk factors.

  1. Page 5, Line 160-168: Sentence should be split into two or three parts.

  1. Air, water, chemical, and noise pollution are considered as the natural environmental risk factors shown in Fig. 2. Table 1, on the other hand, only showed pollution in the form of air and water. What about the other environmental risk factors?

  1. The N-K Model was used to assess the social stability risks associated with major projects. What about the other exiting models (i.e. AHP, Fuzzy, etc. see Miao et al., 2019, https://doi.org/10.1155/2019/2452895; Pan et al., 2019, https://doi.org/10.1155/2019/5783938, Zhang et al., 2019, https://doi.org/10.1111/risa.13311, ). A section discussing this topic would be expected.

  1. Conclusion: Natural environmental risk factors were found to have a negligible impact on a major project. Why? It should be discussed in detail here.

Author Response

(The authors gave the same response as above.)

Round 2

Reviewer 2 Report

The authors do not have answered properly to the comments... 

  1. The authors need to include references for explaining the method and the other techniques in the literature review for social stability risk assessment. They need to justify based on the literature review why they selected those “external uncertain factors”, why they defined that bow-tie model, why they chose the indicators at the program level. They do not use any reference along the sections 2, to identify the risk factors, and section 3, to explain the method.
  2. The knowledge gap is lacking in the document.
  3. The authors do not discuss the results based on the literature.

Author Response

Dear  Reviewer:

Thank you for your letter and for the reviewers’ comments concerning our manuscript entitled “Risk Coupling Evaluation of Social Stability of Major Engineering Based on N-K Model”. Those comments are all valuable and very helpful for revising and improving our paper, as well as the important guiding significance to our researches. We have studied comments carefully and have made correction which we hope meet with approval. Revised portion are marked in red in the paper. The main corrections in the paper and the responds to the reviewer’s comments are as attachment.

Reviewer 3 Report

The authors have addressed the comments raised by the reviewers in a satisfactory manner, therefore, I recommend the manuscript can be published in its present form.

Author Response

Dear Editors and Reviewers:

Thank you for your letter and for the reviewers’ comments concerning our manuscript entitled “Risk Coupling Evaluation of Social Stability of Major Engineering Based on N-K Model”. Those comments are all valuable and very helpful for revising and improving our paper, as well as the important guiding significance to our researches. We have studied comments carefully and have made correction which we hope meet with approval. Revised portion are marked in red in the paper. The main corrections in the paper and the responds to the reviewer’s comments are in the attachment.

Round 3

Reviewer 2 Report

I think that the authors have made an important effort to answer every comment. I consider that the paper can be accepted.